# Antizyme Inhibitor 2-Deficient Mice Exhibit Altered Brain Polyamine Levels and Reduced Locomotor Activity

**DOI:** 10.3390/biom13010014

**Published:** 2022-12-21

**Authors:** Ana Lambertos, Maria Angeles Nuñez-Sanchez, Carlos López-García, Andrés Joaquín López-Contreras, Bruno Ramos-Molina, Rafael Peñafiel

**Affiliations:** 1Department of Biochemistry and Molecular Biology B and Immunology, School of Medicine, University of Murcia, 30120 Murcia, Spain; 2Obesity and Metabolism Laboratory, Biomedical Research Institute of Murcia (IMIB), 30120 Murcia, Spain; 3Airway Disease Section, National Heart and Lung Institute, Imperial College London, London SW3 6LY, UK; 4Centro Andaluz de Biología Molecular y Medicina Regenerativa (CABIMER), Consejo Superior de Investigaciones Científicas (CSIC), 41013 Seville, Spain

**Keywords:** AZIN2, polyamines, mouse brain, gene expression, motoneurons

## Abstract

Background: Alterations in the neural polyamine system are known to be associated with different brain pathological conditions. In addition, the regulation of enzymes involved in polyamine metabolism such as ornithine decarboxylase (ODC), antizymes (AZs), and antizyme inhibitors (AZINs) is critical during brain development. However, while most studies focus on ODC and AZs, less is known about AZIN expression and function in the brain. Thus, our aim was to analyze the expression pattern of AZIN2 during postnatal development, its brain distribution, and its possible implication in phenotypical alterations. Methods: The expression pattern of *Azin2* and other genes related to polyamine metabolism was analyzed by RT-qPCR. β-D-galactosidase staining was used to determine the anatomical distribution of AZIN2 in a Azin2 knockout model containing the βGeo marker. Brain polyamine content was determined by HPLC. The Rota-Rod and Pole functional tests were used to evaluate motor skills in Azin2-lacking mice. Results: Our results showed that expression of genes codifying for AZs and AZINs showed a similar increasing pattern over time that coincided with a decrease in ODC activity and putrescine levels. The analysis of AZIN2 distribution demonstrated that it is strongly expressed in the cerebellum and distributed along the neuron body and dendrites. The ablation of Azin2 showed a decrease in putrescine levels and is related to reduced motor skills. Conclusions: Our study revealed that AZIN2 expression in the brain is particularly limited to the cerebellum. In addition, the ablation of Azin2 leads to a reduction in putrescine that relates to alterations in motor function, suggesting the role of AZIN2 in the functioning of dopaminergic neurons.

## 1. Introduction

The natural polyamines putrescine, spermidine, and spermine are present in all mammalian tissues, where they exert a number of metabolic, cellular, and neurophysiological effects [1]. In the brain, apart from their influence on nucleic acids metabolism, some of their actions on the neurons are known to be related to the regulation of different cation channels such as the N-methyl D-aspartate (NMDA)-type and α-amino-3-hydroxy-5-methyl-4-isoxazole propionic acid (AMPA) glutamate receptors [2,3], the nicotinic acetylcholine receptor [4], inward rectifier K^+^ channels [5], or acid sensing ion channels [6]. Alterations in the neural polyamine system are involved in several brain pathological conditions such as ischemic injury, Alzheimer’s disease, brain tumors, and various types of mental disorders [7,8,9,10,11].

The intracellular levels of polyamines are tightly controlled, mainly through the regulation of their biosynthesis and uptake [1]. Ornithine decarboxylase (ODC) is a key enzyme in the biosynthesis of polyamines that catalyzes the first step of the process, consisting of the transformation of the amino acid L-ornithine into putrescine, a diamine that is used for the synthesis of spermidine and spermine. ODC is precisely regulated by mechanisms acting at the transcriptional, translational, and post-translational levels [12]. Among their direct regulators are antizymes (AZs), which are proteins that interact with ODC, and not only inhibit the enzyme but also stimulate its degradation by the proteasome through a process that is—unusually—independent of ubiquitination [12]. In addition, AZs inhibit cellular polyamine uptake by mechanism that has yet to be identified [13]. Interestingly, the synthesis and degradation of AZs are both affected by the intracellular levels of polyamines by means of a series of particular mechanisms, including ribosomal frameshifting on the AZ mRNA [14]. Furthermore, the effects of AZs can be abrogated by the presence of the antizyme inhibitors (AZINs), proteins that are homologous to ODC, but that lack enzymatic activity [15].

Many studies have demonstrated the existence of ODC in the brain of adult rats and mice, as well as important changes in its activity during brain development [16,17,18]. However, less is known about the expression and function of their regulatory proteins AZs and AZINs in the brain. Antizyme activity and expression of AZ mRNA have been detected in mouse and rat brains [18,19], but the proportion among the three AZs in this tissue is mostly unknown. Regarding AZINs, whereas AZIN1 is present ubiquitously in mammalian tissues, AZIN2 appears to be expressed mainly in the brain and testes [15,19]. In addition, few studies have been devoted to the expression of AZs and AZINs in the different brain regions. In situ hybridization and immunocytochemical analyses revealed intense AZ expression in rat motoneurons [20]. In the rat hypothalamus, the similar expression pattern of ODC and AZIN1 with that of gonadotropin releasing hormone 1 (GnRH1) suggested that this hypothalamic hormone is under the control of polyamines [21]. Increased mRNA expression of *Azin1* was also detected in the rat hypothalamus in response to stressful conditions [22]. Immunohistochemical studies of normal human brain samples detected the presence of AZIN2 mostly in both the white and gray matter and increased accumulation of AZIN2 in brains affected by Alzheimer’s disease [23,24]. However, little is known about the specific role of AZIN2 on brain function. Although initial studies [25] suggest that the protein is responsible for the formation of agmatine, an amine produced by decarboxylation of arginine having different neurological effects, different studies revealed that AZIN2 is devoid of decarboxylating activity [26,27,28].

In the present work, we have compared, by measuring mRNA levels, the expression of *Azin2* with that of other genes related to polyamine metabolism in the brain of adult mice, as well as their changes during the postnatal development of the brain. In addition, we have used a transgenic mice strain that carries a truncated *Azin2* gene fused to the bacterial *lacZ* gene (coding for β-D-galactosidase) under control of the *Azin2* promoter to perform a more detailed analysis of the anatomical distribution of AZIN2 in the whole mouse brain. This mouse model also allowed us to analyze possible phenotypical alterations related to the absence of functional AZIN2 activity in order to better understand the physiological role of this protein in the brain.

## 2. Materials and Methods

### 2.1. Animals

Swiss CD1 and mice with mixed C57BL/6 background were obtained from the Service of Laboratory Animals (University of Murcia) and the Spanish Centre for Cancer Research (CNIO, Madrid, Spain), respectively. Animals were housed in a controlled environment (22 °C ambient temperature and 50% relative humidity under a controlled 12:12 h light-dark cycle, and food and water ad libitum) in standard conditions at the Service of Laboratory Animals (University of Murcia). On the day of tissue collection, mice were anesthetized with sodium pentobarbital and then euthanized by cervical dislocation. The brains were quickly removed, weighed, and processed according to the corresponding procedure. 

### 2.2. Quantitative Real-Time RT-qPCR

Brain tissues were homogenized in Trizol (Ambion Inc, Austin, TX, USA) using a Polytron CH-6010 homogenizer (Brinkmann, Kriens-Luzern, Switzerland) pre-treated with RNAse ZAP (Ambion Inc., Austin, TX, USA). After homogenization, 1 mL of the extract was placed in a 1 mL tube and centrifuged at 12,000× *g* for 1 min at room temperature. Then, the supernatant was diluted with the same volume of 100% ethanol. The mixture was then used to extract total RNA using the PureLink RNA Mini Kit Ambion Inc., Austin, TX, USA) following the manufacturer’s instructions. RNA was eluted in RNAse-free water and checked for concentration and purity using the Nanodrop spectrophotometer system (ND-100 3.3. Nanodrop Technologies, Wilmington, DE, USA). Only samples with a ratio Abs260/Abs280 between 1.8 and 2.1 were used for gene expression assays. cDNA was generated from 5 µg of total RNA using MMLV-Reverse Transcriptase (Sigma-Aldrich, St. Louise, MO, USA) according to the manufacturer’s instructions. Real-time qPCR (RT-qPCR) amplification was performed using the SYBR^®^ Green PCR Master Mix (Applied Biosystems, Buckinghamshire, UK) using a Fast 7500 Real-Time instrument (Applied Biosystems, Buckinghamshire, UK). Expression levels were normalized to the *Actb* housekeeping gene and relative expression was calculated using the 2^−ΔCt^ method. The sequences of the primers used in this study can be found in Appendix A.

### 2.3. Ornithine Decarboxylase Activity Analysis

Brains were homogenized by means of a Polytron homogenizer in 10 mM Tris-HCl, 0.25 M saccharose, 0.1 mM pyridoxal phosphate, 0.2 mM EDTA, and 1 mM (buffer A). The extract was then centrifuged at 14,000× *g* for 20 min, and a fraction of the supernatant was taken to a final volume of 50 μL with buffer A. Decarboxylating activity was determined by measuring ^14^CO_2_ released from L-[1-14C] ornithine (American Radiolabeled Chemicals Inc., St. Louse, MO, USA). The reaction was performed in polypropylene tubes with a tightly closed rubber stopper, hanging from the stoppers two disks of filter paper wetted in a solution of benzethonium hydroxide and methanol (1:1). The samples were incubated at 37 °C from 60 to 120 min, and the reaction was stopped by adding 0.5 mL of 2 M citric acid. The filter paper disks were transferred to scintillation vials with 1.5 mL of scintillation liquid and dpms were counted in a Tri-Carb 2900TR scintillation counter (Perkin Elmer, Waltham, MA, USA).

### 2.4. Generation and Genotyping of Transgenic Mice

Transgenic *Azin2* knock-out (KO) mice were generated as previously described by Lopez-Garcia et al., [29]. Briefly, an ES cells recombinant clone of C57BL/6 background, carrying a gene-trap cassette (Clone IST2418H6, Mouse Accession NM_172875), between exons 4 and 5 of the AZIN2 locus, was generated at the Texas A&M Institute of Genomic Medicine, (http://www.tigm.org, accessed on 24 November 2022) by retroviral insertion. The gene trap cassette includes the following elements: 5′ and 3′ flanking long terminal repeats, splicing acceptor (SA), βGeo marker (βGal and Neo fusion), and a polyadenylation site (Appendix A). ES cells microinjection and selection of chimeras for germ line transmission were carried out at the Animal Facility at the Spanish Centre for Cancer Research (CNIO, Madrid, Spain) resulting in viable heterozygous mice. Genomic DNA was extracted from tail biopsies and genotyping PCR was performed with RedTaq DNA-polymerase (Sigma-Aldrich, St. Louise, MO, USA) according to the manufacturer’s instructions. The recombinant allele (*Azin2*^βGeo^) was amplified using *Azin2* forward primer (5′-GAGGAGTCACATCACCACACG-3′) and V76 reverse primer (5′-CCAATAAACCCTCTTGCAGTTGC-3′). The wild type (WT) allele was amplified using the *Azin2* forward primer mentioned above in combination with the *Azin2* reverse primer (5′-GCTTCATGGTAGACATATGC-3′).

### 2.5. Analysis of β-D-Galactosidase Activity

For β-galactosidase analysis, the brains of WT and KO mice were homogenized by means of a Polytron homogenizer in 50 mM TRIS-HCl pH 7.4 containing 1 mM EDTA and 1% Igepal. Tissue homogenates were centrifuged at 12,000× *g* for 20 min and β-galactosidase activity was determined in the supernatant by measuring the rate of hydrolysis of the substrate o-nitrophenyl-β-D-galactoside (ONPG). The incubations were performed at 37 °C for 30–60 min in 100 mM sodium phosphate buffer (pH = 7.4), 2 mM MgCl_2_, 50 mM β-mercaptoethanol, and 0.66 mg/mL of ONPG, in a total volume of 0.3 mL, and the reaction was stopped by adding 0.6 mL of 500 mM sodium carbonate. Additionally, incubations were also performed in different buffers ranging pH values from 4 to 8, to discard endogenous β-D-galactosidase activity in transgenic extracts. After centrifugation at 12,000× *g* for 5 min, 420 nm absorbance (A_420_) was measured and the activity was expressed as the increase in A_420_ per h and g of wet tissue. 

### 2.6. β-D-Galactosidase Reporter Staining and Immunocytochemistry

Brains dedicated to whole mount detection of the reporter activity were frozen for 20 min after dissection. Next, brains were divided into thick fragments and fixed with 4% PFA in PBS (pH 7.4) for 20 min at room temperature. Tissues were then washed with Tris 0.1 M pH 7.3 and incubated until staining development with a staining solution, containing 1% Xgal (5-bromo-4-chloro-3-indolyl-galactopyranoside) in diethylformamide, Tris 10 mM pH 7.3, 2 mM MgCl_2_, 5 mM potassium hexacyanoferrate (III) and 5 mM potassium hexacyanoferrate (II) trihydrate, 0.005% deoxycholic acid, and 0.01% Nonidet P40.

For histochemical and immunohistochemical analysis of brain sections, mice were standardly perfused with 4% buffered paraformaldehyde. Then, fixed brains were transferred into 20% sucrose in PBS for 48 h for cryopreservation. Vibratome sections of 120 µm-thick from 6% agarose blocks were incubated with Xgal staining solution in a humidity-controlled incubator overnight and counterstained with Neutral Red. Some sections underwent immunocytochemical detection of tyrosine hydroxylase (1:1000) (Cat# NB300-109, RRID:AB_10077691, Novus Biologicals, Centennial, CO, USA), parvalbumin (1:2000) (Cat# P3088 RRID:AB_477329, Sigma-Aldrich, St. Louis, MO, USA), calbindin (1:1500) (Cat# CB 38 RRID:AB_10000340, Burgdorf, Switzerland), and calretinin (1:1500) (Cat# CR 7697 RRID:AB_2619710, Burgdorf, Switzerland) for the characterization of positive cells, tracts, and neuropiles. Finally, sections were mounted by standard procedures using a mounting medium from Dako (Agilent, Santa Clara, CA, USA) and examined with a Leica DMIL (Leica, Wetzlar, Germany) inverted microscope.

### 2.7. Polyamine Analysis

Brains were homogenized in 0.4 M perchloric acid (1:5 or 1:10 *w*/*v*), and after centrifugation at 10,000× *g* for 5 min, the polyamines from the supernatant were dansylated. For this purpose, 100 mL of supernatant were mixed with 200 mL of saturated sodium carbonate and 400 mL of dansyl chloride (10 mg/mL in acetone) and incubated for 2 h at 60 °C. Dansylated polyamines were extracted with toluene and separated by HPLC using a Bondapak C18 column (4.6 × 300 mm; Waters Corporation, Milford, MA, USA) and acetonitrile/water mixtures (running from 70:30 to 96:4 during 30 min of analysis) as mobile phase and at a flow rate of 1 mL/min. 1,7-Diaminoheptane was used as the internal standard, whereas standard solutions of putrescine, spermidine, and spermine were used to calibrate the column. Detection of the derivatives was achieved using a fluorescence detector, with a 340-nm excitation filter and a 435-nm emission filter.

### 2.8. Evaluation of Motor Skills

Rota-Rod 47600 (Ugo Basile, Gemonio, Italy) was used to assess motor coordination in mice. Before the testing day, mice were placed on a stationary rod for habituation to the apparatus followed by two training sessions at testing speed. Rotation speed was fixed at 30 rpm and the time that each mouse remained on the rod was recorded as the score, with a maximum duration of 300 s per trial. On the testing day, three consecutive trails were performed, and the average of the times was recorded. 

Motor behaviour was also evaluated with the pole test. Mice were placed head-up on top of a vertical wooden pole (40 cm long, 0.8 cm in diameter) with the base of the pole placed in the home cage. Once placed on the pole, the time that every mouse took to orient downward and descend the pole back into cage was recorded. All mice received two training sessions consisting of 5 trials each. On the testing day, five consecutive trials were performed, and the average of the times was recorded.

### 2.9. Statistical Analysis

Statistical significance was determined by ANOVA, followed by the post hoc Newman–Keuls multiple range test, or by the Student’s *t*-test using the GraphPad Prism 8.0.1 software. Differences with a *p*-value < 0.05 were considered significant.

## 3. Results

### 3.1. mRNA Expression Levels of Azin2 and Related Genes in the Mouse Brain

The expression of *Azin2* and its paralogues *Azin1* and *Odc*, as well as those of the two antizymes *Az1* and *Az2*, were studied in the mouse brain. For that purpose, mRNA levels were measured by quantitative RT-qPCR in the whole brain of mice of different ages, from the postnatal to the adult periods. The levels of *Azin2* mRNA expression remained low during the first postnatal week followed by a sudden raise during the second week (Figure 1a). After week three, mRNA expression levels of *Azin2* remained increasing, although at a slower rate, until the end of the experiment (Figure 1a). On the other hand, expression levels of *Azin1* continuously increased during the first three weeks, reaching values similar to those found in 60-day-old mice on day 20 (Figure 1b). The analysis of *Az1* and *Az2* mRNA levels revealed that there was a linear increase during the three postnatal weeks, reaching levels close to those found in adult animals on day 20 (Figure 1c,d). That is, the expression of *Az1* was around 10 times higher than *Az2*. No expression of *Az3* could be detected. As shown by Figure 1e, the expression profile of *Odc* was quite different to those of its two paralogs. In fact, *Odc* mRNA content decreased along the two first weeks, and increased thereafter. However, the decrease of ODC activity in this period was more pronounced than that found in *Odc* mRNA. Interestingly, in contrast to *Odc* mRNA expression, such decarboxylating activity did not increase after the third postnatal week and remained low until day 60, presenting levels approximately 75 times lower than those detected on day 1 (5.864 ± 0.635 mmol ^14^CO_2_ h^−1^ g^−1^ on day 1 vs. 0.078 ± 0.012 ^14^CO_2_ h^−1^ g^−1^ on day 60) (Figure 1f; Appendix A). Accordingly, the decrease in brain ODC activity during the first postnatal weeks was associated with a marked reduction in putrescine concentration that, however, did not affect significantly to the content of spermidine or spermine (Figure 1g).

### 3.2. Azin2-lacZ Expression Pattern in the Mouse Brain

Having observed that the expression of *Azin2* mRNA is regulated during brain development, we next evaluated the brain regions as well as the specific cells where AZIN2 is expressed. We used a transgenic mouse expressing a truncated *Azin2* gene fused to the bacterial *lacZ* gene (coding for β-D-galactosidase), under the control of the endogenous promoter of *Azin2* [29]. This model allows a specific and sensitive detection of the expression of the gene in different brain regions.

To validate this model, we first analyzed the expression by RT-qPCR of *Azin2* and the reporter *lacZ* in the brain of homozygous *Azin2* hypomorphic mice (*Azin2*^βGeo/βGeo^), heterozygous (*Azin2*^+/βGeo^), and wild type mice (*Azin2*^+/+^). As shown in Figure 2a, the levels of *Azin2* mRNA in heterozygous mice (HT) was about 50% of control values in WT, whereas only residual expression (3.7% referred to WT levels) was detected in KO mice. In contrast to *Azin2*, the expression of the reporter gene *lacZ* was highest in KO mice, doubling the value in HT brains. As expected, no expression of *lacZ* was detected in the brain of WT mice (Figure 2b).

Next, we quantified the β-galactosidase activity derived from *lacZ* expression in KO and WT brain extracts by spectrophotometric assay based on the hydrolysis of the substrate ONPG. Whereas bacterial β-galactosidase exerts its activity at neutral pH, some mammalian tissues/other organisms have been reported to express a lysosomal β-galactosidase activity with acidic optimal pH 4.0–4.5 [29,30]. To discard a possible interference between both enzymes, the activity was measured at a wide range of pH values. Figure 2c shows a remarkable β-galactosidase activity in the brain of KO mice detected at neutral pH, in agreement with mRNA levels of *lacZ*. In addition, no activity was detected at acidic pH values neither in WT nor KO brains, consistent with the absence of mammalian β-galactosidase activity in this tissue. These results exhibited a good correlation between neutral β-galactosidase activity of the recombinant protein (AZIN2-βgeo) in KO brains and *Azin2* mRNA levels in WT brains, and therefore evidenced the suitable use of the first as a reporter of the expression of AZIN2 protein.

The activity of β-galactosidase is also evidenced by the substrate Xgal (5-bromo-4-chloro-3-indolyl-β-D-galactopyranoside), whose hydrolysis and later oxidation generate a blue precipitate and thus allows the identification of cells expressing the recombinant protein. In order to detect the areas where AZIN2 is expressed in mice brains, we performed a whole mount histological assay of thick coronal sections/coronal hemisections of WT and KO brains with Xgal (Figure 2d). Whereas WT/control brains remained colourless, discarding nonenzymatic hydrolysis of the substrate/unspecific staining, KO brains developed a robust staining in definite areas of the brain, revealing specific regional expression of AZIN2 in the central nervous system. A more precise analysis of histological in brain sections from KO mice showed that AZIN2 is strongly expressed along the cerebellum cortex and the hippocampus (Appendix A).

Regarding the distribution of AZIN2 at cellular level, the magnification of the images demonstrated that AZIN2 is distributed along the neuron body and dendrites (Figure 3a,b). Furthermore, counterstaining of 4 μm brain sections showed a vesicular accumulation of AZIN2 within the soma (Figure 3c,d).

### 3.3. Influence of AZIN2 Ablation on Polyamine Metabolism in Brain Tissues

In order to confirm the results obtained in the histochemical analysis, we next compared the β-galactosidase activity in separate extracts of cerebellum and whole brain of KO mice as well as *Azin2* mRNA expression levels in WT mice. Our results showed a predominance of enzymatic activity in the cerebellum, while observed values in the whole brain were nearly 50% lower (Figure 4a). Consequently, similar results were obtained for *Azin2* mRNA levels in WT mice, as the values of expression levels in the cerebellum were nearly two-fold higher than in the whole brain (Figure 4b).

To assess whether the expression of AZIN2 affects polyamines distribution, we analysed the amount of putrescine, spermidine and spermine in the cerebellum and whole brain of WT and KO mice by HPLC. Overall, our results showed that the levels of total polyamines detected in cerebellum where about 30% higher than in the whole brain (*p* = 0.038), with 612 ± 86 nmol/g and 423 ± 64 nmol/g, respectively. However, in KO mice, we did not find statistically significant differences in total polyamines levels between the cerebellum and the whole brain (563 ± 33 nmol/g and 478 ± 50 nmol/g, respectively). At the tissue level, spermine showed a moderate yet significant increase in whole brains of KO mice compared to the control (*p* = 0.038), while no differences were detected for total polyamines, putrescine, and spermidine (Figure 4b). Conversely, in the cerebellum, there were no differences in total polyamines, spermidine, or spermine between the groups. Interestingly, the levels of putrescine were significantly reduced (Figure 4c).

### 3.4. Functional Analysis of AZIN2 in Brain: Influence on Motor Activity

To investigate the significance of AZIN2 in specific motor nuclei of the cerebellum, the Rotarod and pole tests were performed. The results from the Rotarod test showed increased fatigue and motor impairment in KO mice reflected by the reduced time to fall compared to the WT group (Table 1). In addition, to assess motor coordination and balance, the pole test was performed. A total of 15 of the 16 mice from the WT group reoriented downwards to descend the pole with a T-turn in under 2 s. Conversely, none of the KO mice were able to turn, sliding down the pole while maintaining the initial orientation (Table 1; Appendix A).

Mice were subjected to the Rota-rod test at a constant speed of 30 rpm for a maximum of 300 s. All values are expressed as mean ± SE. Statistical analysis were performed using the Student’s *t*-test, considering *p* < 0.05 to be statistically significant.

## 4. Discussions

In the present study, we have analyzed the spatiotemporal expression of AZIN2 in the mouse brain during postnatal development. In the first part of this study, we have analyzed the expression patterns of *Azin2*, *Azin1*, *Az1*, *Az2*, and *Odc* mRNA. After birth, the neural system undergoes a rapid development which is characterized by changes in synaptic connectivity and gene expression, among others [31,32]. Our results showed that the expression levels of genes involved in polyamine metabolism and regulation of ODC enzymatic activity follow a development-dependent pattern. Interestingly, while levels of *Azin1* and *Az1* mRNA were increased from postnatal day two, such increases were delayed to day nine for the expression of *Azin2* and *Az2*. Furthermore, levels of *Azin2* showed a slight decrease between days two and five. In addition, ODC activity showed a rapid decrease during the first ten postnatal days, reaching stable levels that can be found in adults by day twenty. On the other hand, the levels of *Odc* mRNA showed a gradual decrease, reaching low levels at day seventeen, followed by an increase from day twenty that remained increasing until day sixty. These results agree with previous studies, where discrepancies between ODC and *Odc* mRNA levels were observed during brain development [17,33], suggesting the post-transcriptional or translational regulation of ODC activity in the brain. Since antizymes negatively affect ODC, the fall in ODC activity found during the postnatal period may be related to the increase in the expression of both *Az1* and *Az2*. However, this possibility appears to come into conflict with the marked increase in the expression of both AZINs that may counterbalance the action of the two antizymes. Concurrently, we analyzed the levels of polyamines at different time-points, i.e., postnatal days five, twenty-five, and thirty-five by HPLC. The reduction in the levels of putrescine after birth concurred with those of ODC activity. However, such decreases were not reflected in the levels of spermidine and spermine. Due to their vital physiological role in mammals, the homeostasis of polyamines levels is tightly regulated at many levels including polyamine biosynthesis, catabolism, and transport [34], which might explain such discrepancies between putrescine levels and the concentration of higher order polyamines.

In order to understand the functional relevance of these enzymes in the brain, we next evaluated the spatial distribution using a KO mouse model. Previous reports have shown that the *Azin1* KO mice are not viable and die soon after birth [35,36]. Thus, we focused on the *Azin2* KO mice previously described by us [29]. Our analysis of the different brain sections showed that AZIN2 is widely distributed, albeit limited to specific areas. Staining with X-Gal in KO mice showed a high expression in cerebellum, hippocampus, and cortical neurons confirming previous observations from our group [37]. Furthermore, previous reports have described the presence of AZIN2 in pyramidal neurons of the cerebral cortex [23] as well as in spinal ganglia, dentate nucleus, and granule cells from the cerebellum [38]. In addition, a more detailed histochemical analysis using X-Gal revealed that the expression of AZIN2 is specific of differentiated cells which shows a vesicular distribution along the axons and the soma which is in line with other studies [38]. In this regard, AZIN2 has been postulated to participate in the regulation of the intracellular vesicle trafficking via regulation of polyamine levels [39]. However, the physiological role of AZIN2 in the central nervous system remains unclear.

The polyamine pathway is known to be involved in several cellular functions such as gene expression, proliferation, or apoptosis [34,40]. In addition, polyamines are known to impact neuronal function, axonal integrity, and cognitive processing [24]. Moreover, research over the last decades has shown the importance of polyamines and AZIN2 in neurodegenerative conditions such as Parkinson disease or Alzheimer’s disease [41,42]. More recently, ODC alterations have been linked to neurodevelopmental disorders [43]. In our model, it would be expected that the absence of AZIN2 would lead to a greater ODC-AZ interaction and thus, decreased levels of polyamines and lower ODC activity. However, previous studies have demonstrated that the increase in levels of putrescine is not necessarily correlated with increased levels of spermine and spermidine [44]. Indeed, although the biosynthesis of polyamines has been described to be restricted to neurons, polyamines might be released and accumulated in astrocytes and other glial cells by uptake, probably via the organic cation transporter 3 [45]. Our results showed that, while levels in putrescine were significantly reduced in cerebellum in KO mice, spermine and spermidine were not affected, suggesting that there are other steps in the metabolism of polyamines crucial for their regulation.

In order to evaluate the consequences of AZIN2 ablation during brain development, we performed different tests to examine cerebellar function. The cerebellum represents a critical region in motor coordination and balance with the cerebral cortex [46]. To the best of our knowledge, this is the first study to evaluate motor deficits in *Azin2* mutant mice during brain development. In the present study, the lack of AZIN2 was related to an alteration in the locomotor function. The rota-rod test showed a significant decrease in latency of fall-off times in the bar in the *Azin2* KO mice. Similarly, *Azin2* KO mice underperformed WT mice in the pole test. Due to the physiological function of AZIN2 as regulator of polyamine concentration and its colocalization with the glutamate receptor, N-methyl D-aspartate (NMDA)-type excitatory glutamate receptors (NMDAR), AZIN2 has been suggested to play a critical role in the regulation of these NMDARs [23]. In this regard, these Ca^2+^ and Na^+^ channels in the cerebellum have specific characteristics that make their function and regulation different from other NMDARs in the rest of the brain. For instance, these receptors have been described to play a special role in the modulation of motor learning and coordination [47]. In addition, NMDARs also have polyamine binding sites that, together with other molecules, regulate their functioning [10]. Thus, considering our results, the relationship between the lack of AZIN2 to an alteration in motor function, suggests the participation of AZIN2 in the correct functioning of dopaminergic neurons, probably regulating the release of neurotransmitters such as dopamine [48]. Finally, it is worth noting that, besides memory and learning, spatial navigation is one of the main functions conferred to the hippocampal region. Albeit that this role seems to be strictly related to the organization of memory [49]. Thus, AZIN2 truncation in this area could contribute to the motor impairment reported in AZIN2 KO mice. However, more behavioral tests used to assess learning and memory in rodents should be performed to assess the role that AZIN2 plays in the hippocampus.

## 5. Conclusions

In conclusion, the results of the present study provide new evidence about the development-dependent gene expression profile for genes involved in polyamine metabolism and regulation in mouse brain. In addition, our detailed histological analyses have revealed that AZIN2 is expressed in the brain, although its distribution is limited to certain areas, particularly in the cerebellum. In addition, we have shown that the ablation of *Azin2* in the brain leads to reduced levels of putrescine and relates to alterations in motor function, suggesting a role of AZIN2 in the functioning of dopaminergic neurons. However, further studies are required to elucidate the exact mechanisms involved in such relationships.

## Figures and Tables

**Figure 1 biomolecules-13-00014-f001:**
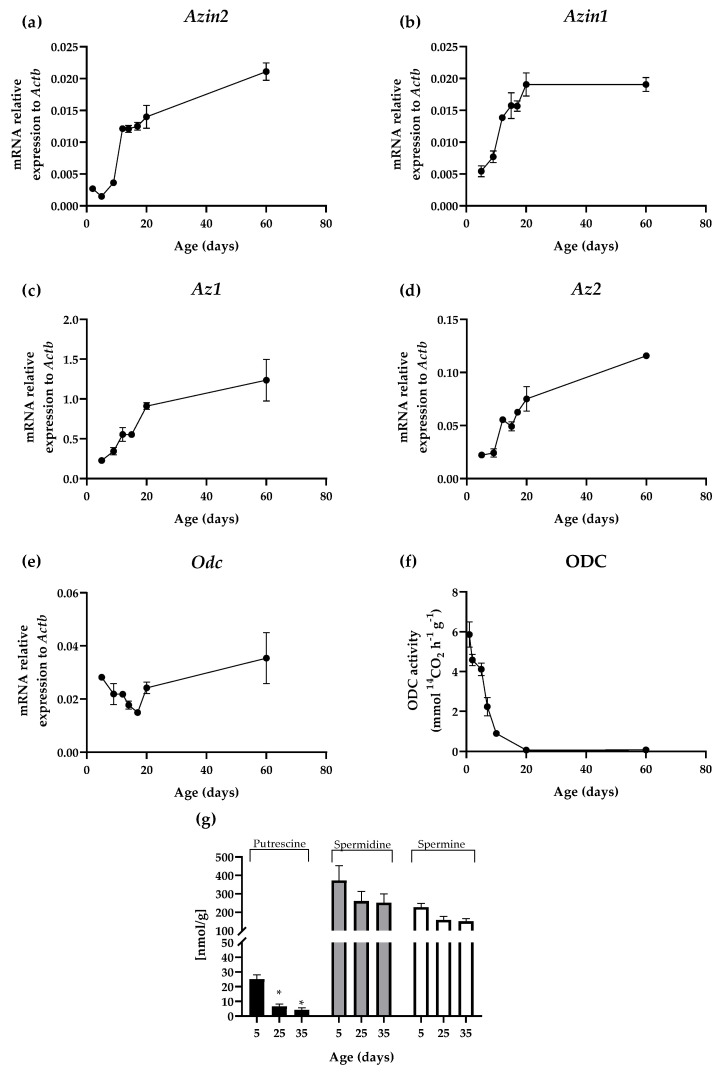
Expression patterns in brain during postnatal period of (**a**) *Azin2*, (**b**) *Azin 1* (**c**) *Az1*, (**d**) *Az2*, and (**e**) *Odc* mRNA expression levels normalized to *Actb* housekeeping gene; (**f**) ODC enzymatic activity expressed in mmol ^14^CO^2^·h^−1^·g^−1^; (**g**) levels of putrescine, spermidine, and spermine expressed in nmol/g of brain. All values are given as mean ± SE. Statistical differences were calculated using the two-way ANOVA followed by the post hoc Newman–Keuls multiple range test. *n* = 6 animals per group. * *p* < 0.05.

**Figure 2 biomolecules-13-00014-f002:**
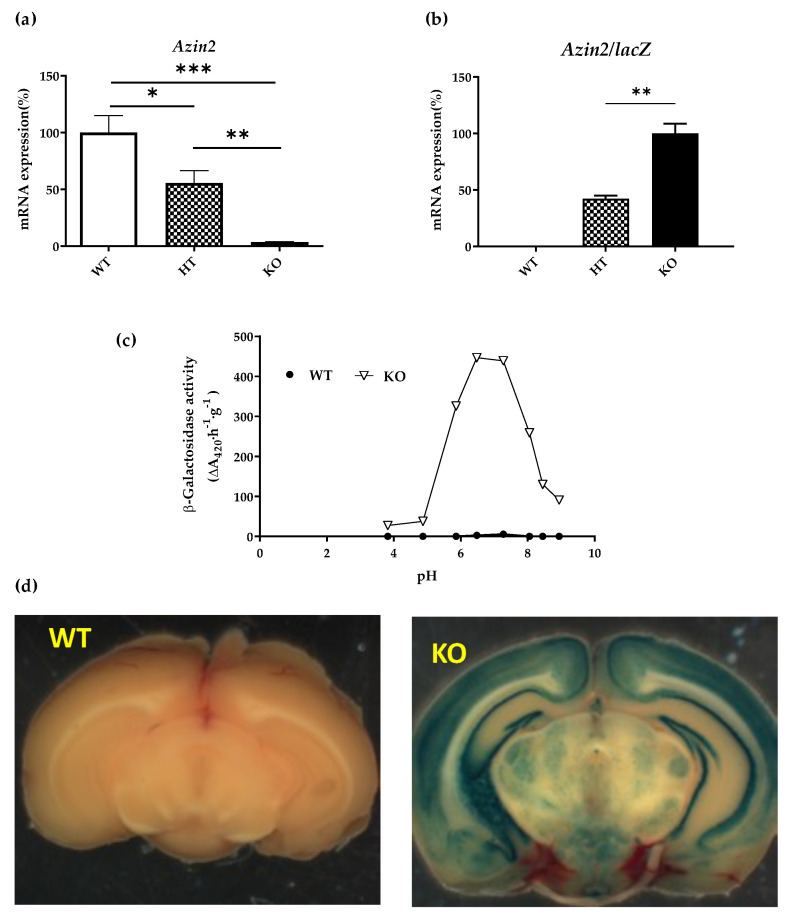
Characterization of transgenic *Azin2* KO mice. (**a**) *Azin2* and (**b**) *Azin2/lacZ* mRNA levels in mouse brains WT, HT and KO adults. (**c**) β-galactosidase activity in KO and WT mice. (**d**) AZIN2 expression in thick coronal sections/coronal hemisections of WT and KO brains with Xgal. All values are given as mean ± SE. Statistical differences were calculated using the two-way ANOVA followed by the post hoc Newman–Keuls multiple range test. *n* = 4 animals per group. (* *p* < 0.05; ** *p* < 0.01; *** *p* < 0.001).

**Figure 3 biomolecules-13-00014-f003:**
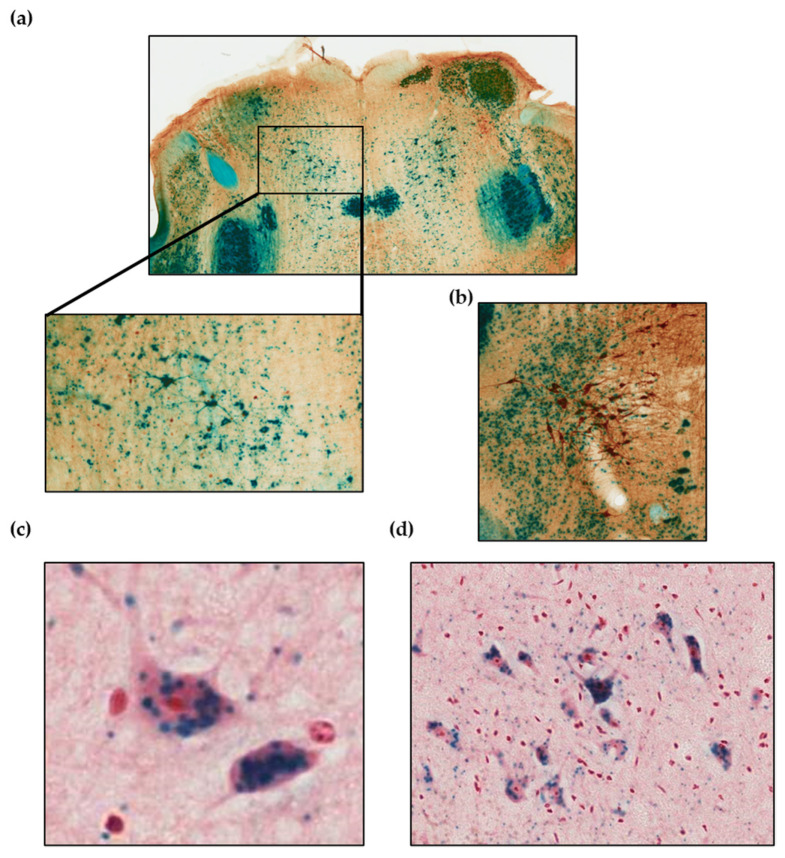
Detection of *lacZ* in 4μm brain sections of KO mice by X-gal staining. (**a**) Magnification showing intense staining of the neuronal body and dendrites. (**b**) Magnification showing neurons that express tyrosine hydroxylase (Th) together with cells expressing AZIN2. In blue β-galactosidase reaction at pH = 7.2; in brown, counter-staining by thyroxine hydroxylase and peroxidase antibodies; (**c**,**d**), neurons from different regions showing accumulation of AZIN2 in the neuronal body. In blue beta galactosidase reaction at pH = 7.2; counter-staining with neutral red.

**Figure 4 biomolecules-13-00014-f004:**
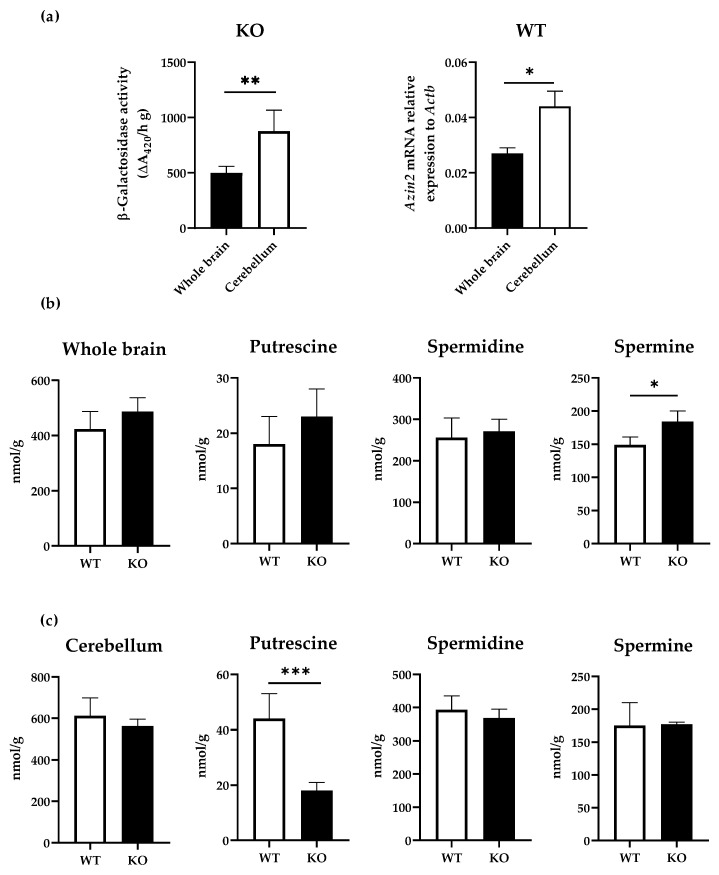
Polyamine levels in brain and cerebellum of KO mice. (**a**) β-galactosidase activity detected in whole brain and cerebellum of KO mice expressed as ΔA420 h^−1^·g^−1^, and *Azin2* mRNA expression levels in whole brain and cerebellum of WT mice normalized to *Actb*. Polyamine levels detected in (**b**) whole brain and (**c**) cerebellum of WT and KO mice expressed as nmol/g. Values are given as mean ± SE. Statistical analysis was performed using the Student’s *t*-test. *n* = 4 animals per group. * *p* < 0.05, ** *p* < 0.01, *** *p* < 0.001.

**Table 1 biomolecules-13-00014-t001:** Rota-rod and Pole locomotor activity tests.

*ROTA-ROD*			
	WT (*n* = 17)	KO (*n* = 17)	*p*-value
Score (cm/min)	107 ± 13	88 ± 24	0.03
Time (s)	313 ± 27	42 ± 36	< 0.0001
*POLE TEST*			
	WT (*n* = 16)	KO (*n* = 16)	
Successful rate	15/16	0/16	
T-turn (s)	1.55 ± 3.2		
T-total (s)	5.6 ± 3.2		

## Data Availability

Not applicable.

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
