# Peer review of "Antizyme Inhibitor 2-Deficient Mice Exhibit Altered Brain Polyamine Levels and Reduced Locomotor Activity"

_biomolecules, 2022, doi:10.3390/biom13010014_

Round 1

Reviewer 1 Report

The manuscript titled “Antizyme inhibitor 2-deficient mice exhibit altered brain polyamine levels and reduced locomotor activity” showed that the ablation of AzIn2 leads to a reduction in putrescine, which is probably involved in alteration in motor function. These results suggest that AZIn2 is involved in the function of dopaminergic neurons. The results are clearly shown, but the followings should be revised.

1.     Abbreviations such as βGeo, GnRH1, ONPG etc. should be collectly shown.

2.     HET is shown in the sentence, and HT is shown in the figure. It is better to use the same abbreviation in both sentence and figure.

3.     Fig. 4 legend: a) whole brain and b) cerebellum b) whole brain and c) cerebellum .

4.     There are no sentences in 5. Conclusions. Conclusions should be described.

Reviewer 2 Report

The authors report the effects of altered Azin2 expression on polyamine levels and behaviour. I have the following comments.

1. The ODC expression and activity after P20 are not concordant as the authors discuss. However, the report a ODC activity of zero after P20. Is this realistic?

2. Why the authors do not follow the fact that also in hippocampus a considerable somatic expression of Azin2 is detectable (only a short note on line 290 shows this, cf. also Allen's brain atlas)? That could influence the interpretation of behavioral data and should be discussed in detail.

3. Fig. 1 and Fig. 4: How many animals were used?

Author Response

Reviewer 2.

Comment 1: The authors report the effects of altered Azin2 expression on polyamine levels and behaviour. I have the following comments.

The ODC expression and activity after P20 are not concordant as the authors discuss. However, the report a ODC activity of zero after P20. Is this realistic?

Response to comment 1: Thanks for your comment. As the reviewer has highlighted there is a high discordance between ODC activity and Odc mRNA expression. As discussed in the manuscript this fact has been also observed by other authors. We are aware that the figure suggests that ODC activity is reduced to 0. However, our analyzes show a reduction in said activity that stabilizes at low levels after 20 days (see table below). We have now included a sentence in the manuscript and the table in supplementary material for clarity purposes.

Table S2. Ornithine carboxylase activity during the postnatal period.

Time

(days)

mmol 14CO2 h-1 g-1

(mean ± SE)

1

5.864 ± 0.635

2

4.583 ± 0.278

5

4.117 ± 0.316

7

2.235 ± 0.452

10

0.905 ± 0.123

20

0.070 ± 0.009

60

0.078 ± 0.012

“Interestingly, in contrast to Odc mRNA expression, such decarboxylating activity did not increase after the third postnatal and remained low until day 60, presenting levels approximately 75 times lower than those detected on day 1 (5.864 ± 0.635 mmol 14CO2 h-1 g-1 on day 1 vs. 0.078 ± 0.012 14CO2 h-1 g-1 on day 60) (Fig. 1f; supplementary Table S2).”

Comment 2: Why the authors do not follow the fact that also in hippocampus a considerable somatic expression of Azin2 is detectable (only a short note on line 290 shows this, cf. also Allen's brain atlas)? That could influence the interpretation of behavioral data and should be discussed in detail.

Response to comment 2: Thanks for highlighting this. We agree that AZIN2 truncation in the hippocampus could influence the motor impairment. We have included a paragraph discussing this fact in the discussion section.

Page 13, lines 425-431:

“Finally, it is worth noting that, besides memory and learning, spatial navigation is one of the main functions conferred to the hippocampal region albeit this role seems to be strictly related to the organization of memory [49]. Thus, AZIN2 truncation in this area could contribute to the motor impairment reported in AZIN2 KO mice. However, more behavioral tests used to assess learning and memory in rodents should be performed to assess the role AZIN2 plays in the hippocampus.”   

Comment 3: Fig. 1 and Fig. 4: How many animals were used?

Response to comment 3: Thanks for the remark. We have now included the number of animals in both figure legends.

“Figure 1. Expression patterns in brain during postnatal period of a) Azin2, b) Azin 1 c) Az1, d) Az2, and e) Odc mRNA expression levels normalized to Actb housekeeping gene; f) ODC enzymatic activity expressed in mmol 14CO2h-1g-1; g) levels of putrescine, spermidine, and spermine expressed in nmol/g of brain. All values are given as mean ± SE. Statistical differences were calculated using the two-way ANOVA followed by the post hoc Newman-Keuls multiple range test. N = 6 animals per group. *p < 0.05. “

“Figure 4. Polyamine levels in brain and cerebellum of KO mice. a) β-galactosidase activity detected in whole brain and cerebellum of KO mice expressed as ΔA420 h-1g-1, and Azin2 mRNA expression levels in whole brain and cerebellum of WT mice normalized to Actb. Polyamine levels detected in b) whole brain and c) cerebellum of WT and KO mice expressed as nmol/g. Values are given as mean ± SE. Statistical analysis were performed using the Student’s t-test. N = 4 animals per group. *p<0.05, **p<0.01, ***p<0.001.”

Round 2

Reviewer 2 Report

The authors have addressed all of my comments.